# Benchmarking of Sensor Configurations and Measurement Sites for Out-of-the-Lab Photoplethysmography

**DOI:** 10.3390/s24010214

**Published:** 2023-12-29

**Authors:** Max Nobre Supelnic, Afonso Fortes Ferreira, Patrícia Justo Bota, Luís Brás-Rosário, Hugo Plácido da Silva

**Affiliations:** 1Department of Bioengineering (DBE), Instituto Superior Técnico (IST), 1049-001 Lisbon, Portugal; patricia.bota@tecnico.ulisboa.pt (P.J.B.); hsilva@lx.it.pt (H.P.d.S.); 2Instituto de Engenharia de Sistemas e Computadores—Microsistemas e Nanotecnologias (INESC MN), 1000-029 Lisbon, Portugal; afonsofortes@tecnico.ulisboa.pt; 3Instituto de Telecomunicações (IT), 1049-001 Lisbon, Portugal; 4Cardiology Department, Santa Maria University Hospital (CHLN), Lisbon Academic Medical Centre, 1649-028 Lisbon, Portugal; lsrosario@medicina.ulisboa.pt; 5Cardiovascular Centre of the University of Lisbon, Lisbon School of Medicine, 1649-028 Lisbon, Portugal

**Keywords:** photoplethysmography, saturation, wave morphology, heart rate, peak detection, measurement sites

## Abstract

Photoplethysmography (PPG) is used for heart-rate monitoring in a variety of contexts and applications due to its versatility and simplicity. These applications, namely studies involving PPG data acquisition during day-to-day activities, require reliable and continuous measurements, which are often performed at the index finger or wrist. However, some PPG sensors are susceptible to saturation, motion artifacts, and discomfort upon their use. In this paper, an off-the-shelf PPG sensor was benchmarked and modified to improve signal saturation. Moreover, this paper explores the feasibility of using an optimized sensor in the lower limb as an alternative measurement site. Data were collected from 28 subjects with ages ranging from 18 to 59 years. To validate the sensors’ performance, signal saturation and quality, wave morphology, performance of automatic systolic peak detection, and heart-rate estimation, were compared. For the upper and lower limb locations, the index finger and the first toe were used as reference locations, respectively. Lowering the amplification stage of the PPG sensor resulted in a significant reduction in signal saturation, from 18% to 0.5%. Systolic peak detection at rest using an automatic algorithm showed a sensitivity and precision of 0.99 each. The posterior wrist and upper arm showed pulse wave morphology correlations of 0.93 and 0.92, respectively. For these locations, peak detection sensitivity and precision were 0.95, 0.94 and 0.89, 0.89, respectively. Overall, the adjusted PPG sensors are a good alternative for obtaining high-quality signals at the fingertips, and for new measurement sites, the posterior pulse and the upper arm allow for high-quality signal extraction.

## 1. Introduction

Cardiovascular diseases are the leading cause of death worldwide, with 17.9 million deaths each year [1,2], and there is a high prevalence of chronic cardiovascular disease that is the major cause of hospital admissions. Since chronic patients are ambulatory, there has been a growing interest in wearable devices capable of monitoring cardiovascular parameters on a daily basis [3]. Wearable devices are becoming a part of our daily routines, ranging from smartwatches to fitness trackers or even smart clothes [4,5,6]. Along with the commodities they offer, health monitoring technology has also branched into these devices, with the latest advances allowing for continuous health monitoring outside clinical settings [7]. At the same time, these devices are becoming increasingly accurate, meeting clinical standards [8,9]. Photoplethysmography is a must-have sensor in wearable devices for heart-rate monitoring, due to its simplicity, versatility, and non-invasiveness. Most often, these sensors are used at the fingertips due to their widely spread and shallow vascular bed [10], and, consequently, high signal amplitudes can be obtained [11]. While in an experimental setting these body parts are preferred, for lifestyle activity data collection they are impractical, and also much more susceptible to motion artifacts [8]. A caveat of PPG sensors is their tendency to saturate, which does not provide meaningful clinical information [12,13]. On top of this, when applying common biosignal filtering techniques, saturated portions lead to distorted filtered signal, a ringing effect, which in turn contributes to inaccurate peak detection and extracted features [12,14,15,16]. Consequently, alternative measurement sites need to be explored and validated to ensure that high-quality signals can be obtained. While many different anatomic regions are being used in research to obtain unsaturated data, a systemic analysis of the accuracy and quality of these new signals has not been made. Moreover, health parameters, such as heart rate (HR) and heart-rate variability (HRV), need to be accurately derived. In this paper, we evaluated the following anatomical sites: for the upper limbs, the anterior upper arm, posterior and anterior wrist, middle finger, and index finger regions; while for the lower limbs, the medial ankle and second left toe.

## 2. Background

### 2.1. Photoplethysmography

Photoplethysmography is a non-invasive method of measuring blood volume changes in the human body. Incident light with a specific wavelength is used to extract optical properties from the microvascular bed of the skin and modulate the pulsatile blood flow [17].

PPG sensors consist of a photoemitter (PE) and a photodetector (PD). The light emitted from the source is absorbed, scattered, and reflected by the human body’s tissues. The PD’s position dictates whether the PPG is transmissive or reflective, as seen in Figure 1. In both of these types, the blood flow is modulated by the amount of light that is transiently absorbed by the skin. Transmissive PPGs have the PD attached across the body part, acquiring a signal based on the attenuation of the light crossing it. Reflective-type PPGs have the PD positioned side by side with the PE. In this case, the obtained signal comes from the scattering and reflection of the light. Transmissive PPGs are capable of more stable readings than reflective PPGs. However, reflectance mode PPG is more easily implemented, especially in wearable devices, since both the PE and the PD can be integrated in close proximity to each other [10,18].

The intensity of the captured light depends on various biological factors, such as tissue opacity of the interposed skin, connective tissue, bone, and also the amount of blood in the capillary vessels. Blood has a higher absorption coefficient than the other bodily components; therefore, variations in blood volume, such as arterial pulsation, account for the measurable cardiac cycle variations observed in the PPG signal [10].

Although different wavelengths have been used for PPG acquisitions, higher wavelengths can reach higher depths of penetration, with the wavelength of 660 nm being able to reach subcutaneous tissues. Red wavelengths, 640–660 nm, and infrared wavelengths, 880–940 nm, are commonly used for PPG readings [19]. Nonetheless, PPG sensors working in these ranges are significantly influenced by heat radiation from human skin. For instance, when comparing PPG signals using these wavelengths upon varying temperatures, the correlation between the signal and the biological feature being studied is weaker [20]. To overcome this, green wavelengths (around 525 nm) can be used, because they are not affected as much by both temperature effects and motion artifacts [19,20,21].

The PPG signal comprises two main components, the pulsatile or cardiac cycle (CCC) and superimposed or basal (BC) components. The pulsatile component originates from the blood volume variations created by the cardiac activity. As such, this signal portion depends on the cardiac phases, namely the systolic and diastolic phases. By isolating a single PPG pulse, four main fiducial points can be seen, as shown in Figure 2.

The foot or onset corresponds to the beginning of the pulse (in the systolic phase), followed by the systolic peak, which is the maximum peak of the pulse signal, occurring after. The rising edge of the signal from onset to peak thus represents the rising blood volume within the measured body area. Next, a depression called the dicrotic notch occurs, as the aortic valve closes, followed by a lower peak, the diastolic peak. This second local maximum represents the reflected Windkessel wave [23,24]. The CCC component can thus be used to estimate cardiac-related features such as heart rate, heart rate variability, and blood pressure. Since these variations are synchronized with the cardiac cycle, they are also related to vasodilation, vasomotor, and vascular tones. The superimposed component, or nonpulsatile component, is mainly influenced by tissue composition and basic blood volume within the measurement site, and is often attenuated through signal conditioning [25]. This component is affected by internal parameters such as breathing, thermoregulation, and the activity of the sympathetic nervous system. Moreover, external parameters such as ambient light can also affect this signal component [10]. Because the amplitude of the PPG signal depends on the aforementioned internal and external factors, its measuring unit is considered arbitrary.

### 2.2. Measuring Sites

The most common measurement site for PPG signals is the finger. Being a peripheral part of the body with high blood perfusion allows for high signal amplitude [26]. The key disadvantage of this site is tied to the susceptibility of the sensor to motion artifacts. During daily activities, the fingers are constantly moving, creating a significant amount of motion noise which, in turn, compromises the measured signal [27]. Other measurement sites have been described in the literature, such as the earlobe, the wrist, the arm, or the ankle.

The earlobe has a much higher blood perfusion than the other measurement sites, even when compared to the finger, making it very attractive for obtaining high amplitude [28]. Earlobe sensors are also very easy to fabricate, but since the most common approach is with a spring-loaded ear clip, their use in long-term recordings becomes uncomfortable for the subject.

The wrist’s main advantages as a measurement site are its ease of use and discreetness. Many sensors are adapted into smartwatches, becoming a part of the subject’s usual garments, not requiring the volunteer to use any other equipment. The signal amplitude is lower than at the finger, but it is also less susceptible to motion artifacts [29,30].

The upper arm is also used despite the lower signal amplitude, due to its proximity to human arteries and its potential to be inserted into clothing, becoming just as comfortable as a wrist sensor [31].

The ankle, while less common, has a significant potential for implementing PPG data collection in discreet form factors. Being one of the farthest extremities of the body from the heart, it allows for the collection of peripheral blood perfusion data more accurately, helping detect conditions such as peripheral artery disease or peripheral vascular disease. It can also be easily integrated into undergarments such as socks to allow for constant monitoring without sacrificing comfort [22,32,33,34].

## 3. Materials and Methods

### 3.1. Data Collection

Two signal acquisition devices were used, consisting of the BITalino (r)evolution Plugged (PLUX, S.A., Lisbon, Portugal) and the ScientISST CORE (https://www.scientisst.com/).

Both these devices have been validated for their use in biosignal acquisitions, including PPG monitoring in research contexts. For instance, the BITalino device has been widely used within the research community for biosignal acquisitions [35,36,37,38,39,40]. Moreover, the ScientISST CORE [41] is a novel signal acquisition board especially developed for biomedical applications, and has also been used in similar research contexts [22,42,43,44,45]. For the BITalino, the 10-bit ADC was used, and for the ScientISST CORE, the 23-bit ADC was used. All data were collected with a sampling rate of 1000 Hz.

Among various PPG sensors available, off-the-shelf reflective-mode sensors were used for PPG acquisition (Pulse Sensor from World Famous Electronics LLC, New York, NY, USA). These sensors were selected for their cost-effectiveness, open-source hardware, and established usage in prior research [46,47,48,49,50,51,52,53,54,55]. As an analog sensor, various strategies can be employed to enhance its performance. These approaches encompass adjustments to the PD sensor, modifications to LED characteristics, changes in transimpedance amplification, and enhancements to signal conditioning.

Within the first strategy, exploring the fine-tuning of sensor sensitivity or adjusting the wavelength between the LED and the PD sensor were potential avenues. The second strategy involved the analysis of LED intensity or wavelength. Both strategies would necessitate either new specific parts or the ability to modify existing circuit components.

Our chosen approach, the third strategy, focused on adjusting transimpedance amplification. This allowed us to modify the final signal gain simply by replacing a specific resistance with another of the desired value.

Alternatively, addressing the saturation issue through signal conditioning would require a restructuring of the circuit’s filtering section to achieve a different outcome. However, this path posed challenges, as increased filtration might not guarantee the elimination of saturation and could potentially introduce artifacts resembling the correct waveform but inaccurately represented.

Given the array of optimization strategies, we opted for the modification of transimpedance amplification due to its significant impact on the output signal in a straightforward manner. This decision streamlines implementation for future researchers, ensuring accessibility and ease of adoption in subsequent studies. As mentioned before, the gain of the amplification stage was tuned, which, by hypothesis, can either reduce signal saturation or increase sensitivity. As shown in Figure 3, the amplification stage of the circuit has a fixed gain, as given by Equation (Equation 1) [56]:(1)$$A_{v}=1+\frac{R6}{R5}$$

With the original resistance values of the sensor (*R6* = 3.3 MOhm and *R5* = 10 kOhm), the gain is 331. The amplification gain can thus be tuned by changing the resistance value of resistor *R6*.

To collect the data, fine-tuned sensors (Figure 4 and Figure 5) were used in different measurement sites, as seen in Table 1.

Moreover, the ScientISST CORE was covered by a 3D-printed enclosure, with an internally connected PPG sensor with G_high (See Table 1). The box has an opening where the PPG sensor is placed, allowing for data collection. To fit the 3D-printed box, a socket was also 3D-printed and sewn into an elastic band (Figure 5).

Data were collected from 28 volunteers in three setups. Due to synchronization issues and low signal quality, only 21 experiments were processed from the first setup, 22 from the second setup, and 20 from the third setup. Subjects were asked for anthropometric data, namely age, weight, and height. (Table 2). For the age, to further protect the volunteer’s privacy, it was asked through 5 different age ranges. As such, the median will be the age range that was most selected.

An experimental protocol was employed to obtain PPG data at rest and while performing an everyday action (writing on a computer). New measurement sites were also tested to explore their potential. Overall, three experimental setups were used (Figure 6).

First Setup

BITalino connected sensors:–G_low sensor on the left hand’s middle finger (G_lowMidBit);–G_def sensor on the left hand’s index finger (G_defIndBit).ScientISST connected sensors:–G_high box sensor on the left arm, three fingers above the anterior elbow (G_highArmSci);–G_high sensor placed above the ulnar styloid process on the left anterior pulse (G_highAntSci).

Second Setup

BITalino connected sensors:–G_low sensor on the left hand’s middle finger (G_lowMidBit);–G_def sensor on the left hand’s index finger (G_defIndBit).ScientISST connected sensors:–G_high box sensor on the left arm, three fingers above the anterior elbow (G_highArmSci);–G_high sensor placed above the ulnar styloid process on the left posterior pulse (G_highPosSci).

Third Setup

BITalino connected sensors:–G_low sensor on the left hand’s middle finger (G_lowMidBit).ScientISST connected sensors:–G_high box sensor on the left leg, above the medial ankle (G_highAnkSci);–G_def sensor placed on the second toe of the left foot (G_defToeBit).

To temporally align the signals acquired by the two devices, optical synchronization was employed, using an LED and a light sensor operating in the visible spectrum. The LED was connected to a digital output pin of the BITalino, and the light sensor (LUX sensor from PLUX, S.A., Lisbon, Portugal) was connected to an analog input channel of the ScientISST. In the experiments, synchronization was performed by triggering a brief pulse on the LED and recording the respective light pulse at the LUX sensor.

Two minutes of at-rest recording was conducted for the three setups. Then, in the first two setups, another two minutes were recorded while the volunteer copied a text on the computer. To more easily address each of the sensors, the abbreviations shown in Table 1 will be used.

### 3.2. Data Processing

In order to establish the upper limit of saturation on the measurement device, we conducted experiments to determine a range of ADC values for each specific fine-tuned sensor. These sensors were tested while connected to the respective acquisition board, as outlined in Table 1. During each experiment, the signal saturation was measured for a duration of 10 seconds (s) by directing the LED light from an additional sensor directly onto the sensor being tested in that particular moment.

We iteratively determined the amplitude threshold for signal saturation by analyzing the statistical results of each test. This process was repeated for each sensor until 10 s of saturation were achieved. To process the obtained signals, a manual threshold was defined to classify each sample as either saturated or not.

To calculate statistical values, we followed a method similar to [12], considering signal saturation at the upper limit as the mean value of the samples above the defined threshold subtracted by one standard deviation (μ−1×σ). This subtraction resulted in a more conservative threshold that included more values of saturation. To prevent the detection of outliers, only one standard deviation was subtracted.

### 3.3. Quality Evaluation

To evaluate the signals, they were initially categorized into two sections: ‘at-rest’ and ‘movement.’ The first 2 min of signals from the first two setups were designated as at-rest periods, while the remaining signal data received the movement label. Signals from the third setup were uniformly labeled as at-rest.

To assess signal quality, we leveraged the periodicity of the signal, employing spectral entropy (*SE*), calculated using the formula based on Shannon’s entropy:(2)$$SE\left(f1,f2\right)=-\frac{\sum_{f1}^{f2}\hat{P}\left(f\right)\log_{2}\left(\hat{P}\left(f\right)\right)}{\log_{2}\left(N\right)}$$

Here, $\hat{P}\left(f\right)$ represents an estimate of the power spectral density at frequency *f*, and *N* is the number of frequency bins in the range [f1,f2]. The *SE* metric yields a value based on the frequency distribution in the analyzed signal portion. High entropy indicates a uniform distribution of power frequency across the spectrum, while low entropy suggests a ’peak’ in the power density spectrum, where the power at a certain frequency is significantly higher than the rest [57]. This metric is translated into a range of 0 to 1, representing pure periodic signals on one end and random noise on the other.

For PPG signals, *SE* serves as a valuable quality indicator, revealing the prominence of the signal’s periodicity. A poor-quality signal is associated with values higher than 0.8 [58,59]. In each signal portion, consecutive 4 s intervals between 1 Hz and 3 Hz were used to calculate *SE* values [60]. The average of these *SE* values was computed per signal, and a final average per sensor was obtained. This process was repeated for non-saturated signals.

A comparison of pulse waves was conducted between sensors to determine their correlation. Signal pairs were created for each experiment. In the first two layouts, the G_lowMidBit signal was paired with all other signals. In the third layout, two pairs were formed: G_lowMidBit with G_defToeSci and G_defToeSci with G_highAnkSci.

Using BioSPPy [61], each signal underwent initial processing through a 4th-order Butterworth bandpass filter with critical frequencies [1 Hz, 8 Hz]. PPG peaks were extracted using Elgendi’s peak detection algorithm [62] with a peak window of 0.18, a beat window of 0.69, and a beat offset of 0.01 [63]. The signals were visually inspected to ensure that the peak waveforms co-occurred in both recordings. The next step involved selecting a pulse wave, normalizing the data, and synchronizing once more (i.e., prior to optical synchronization) to account for small delays arising from different measurement sites. The following correlation features were calculated:Pearson Correlation Coefficient (*PCC*)—*PCC* is a statistical measure of the degree of linear correlation between two variables calculated as:
(3)$$PCC\left(x,y\right)=\frac{cov(x,y)}{\sigma_{x}*\sigma_{y}}$$
with cov(x,y) being the covariance of two variables and $\sigma_{x}$ the standard deviation of the variable [64].Cosine Similarity (*CS*)—*CS* is a measure of similarity between two vectors obtained with:
(4)$$CS\left(x,y\right)=\frac{x\cdot y}{∥x∥∥y∥}$$Normalized Euclidean Distance (*nED*)—*nED* is the normalized distance between two points, computed as:
(5)$$nED\left(x^{\prime },y^{\prime }\right)=\sqrt{{\left(x^{\prime }-y^{\prime }\right)}^{2}}$$
with $x^{\prime }$ and $y^{\prime }$ being the normalized versions of signals *x* and *y*. The signals were normalized using min–max scaling, with 0 as the minimum value and 1 as the maximum value.

To assess automatic peak detection in the signals collected by each sensor in every experiment, a ground truth was established through manual annotation of the systolic peaks [65]. In the first and second setups, the G_lowMidBit signal was utilized, while in the third setup, G_defToeSci was employed. G_lowMidBit was chosen because the sensor with default parameters often saturated in systolic peak regions. G_defToeSci was selected as it closely replicates the anatomical structure of the index finger (the standard location for PPG acquisition) and demonstrated higher signal amplitude than the ankle. Subsequently, the respective ground truth was compared to the automatically detected peaks in each signal.

A true positive was recorded if a ground-truth peak had a corresponding automatic peak within a 50-millisecond range. Conversely, if no automatic peak existed within that range, it was classified as a false negative. All automatic peaks without a corresponding ground-truth peak were deemed false positives. Using these values from the confusion matrix, the sensitivity and precision of the automatic peak detector were calculated.

For each signal, the HR was estimated based on the temporal differences between consecutive systolic peaks, measured in Beats Per Minute (BPM). This metric was calculated from both the automatic peaks and the manual ones (ground truth). Additionally, we assessed the statistical variability of HR, including the mean and standard deviation.

Next, we determined the difference between the mean HR computed from the automatic peaks and the ground-truth peaks. These differences were then averaged across all subjects and layouts for each sensor, providing the statistical variability of the HR estimation error within the test population. This variability was characterized by the mean and standard deviation of the differences in mean HRs per sensor [22].

Alongside the mean HR difference, the percentual error was also calculated through the following formula [66]:(6)$$Error=\frac{AutomaticHR-GroundTruthHR}{GroundTruthHR}$$

The peak-to-peak intervals were also used to plot Poincaré plots.

## 4. Results

### 4.1. Saturation

After conducting the saturation experiment, the results in Table 3 indicate a notable variance between the minimum saturation values of each sensor and the corresponding mean and median. This disparity underscores that saturation is not an instantaneous process. Consequently, employing statistical values is justified for determining the final saturation threshold, rather than relying on a simple manual approach.

By applying these updated thresholds to the collected signals, we calculated the mean saturation percentage for each measurement site, as shown in Table 4. As anticipated, the original sensor, G_defIndBit, exhibited high saturation at 18.554%. This aligns with the observed pronounced signal saturation during systolic peaks, where the amplitude increased within the defined saturation range.

On the other hand, the lower gain sensor, G_lowMidBit, while not completely eliminating saturation, demonstrated a significant improvement in signal quality, with a mean saturation of only 0.479%. An illustrative example is presented in Figure 7.

The sensors used in the new measurement sites, namely the G_high sensors that were applied to the wrist, arm, and ankle, have almost no saturation, since for these locations, the signal amplitude was dampened (lower sensitivity).

### 4.2. Signal Quality

After calculating the mean Spectral Entropies (mSEs) for each sensor during both signal periods (refer to Table 5 and Table 6), all signals exhibited high quality, with mSEs consistently below 0.8. Notably, sensors on the fingers displayed the lowest mSEs, closely followed by the posterior wrist and arm sensors. An unexpected outcome emerged with the G_defIndBit sensor registering the lowest mSE. This anomaly can be attributed to saturation, resulting in reduced entropy due to continuous constant values.

To validate this proposition, examining the mSE of the non-saturated signal revealed an increase for G_defIndBit from 0.572 to 0.605. Although it approached the other mSEs, it remained the lowest value. The mSEs obtained during movement periods exhibited slightly higher values, yet still below the 0.8 quality threshold. Swarm plots were generated for both at-rest (see Figure 8) and movement (see Figure 9) periods.

### 4.3. Signal Correlation

To assess wave morphology, signal pairs were assigned (refer to Table 7 and Table 8). Examining the resting signal portions (see Table 7), the pairing of G_defIndBit with G_lowMidBit reveals that the sensor gain change successfully maintained wave morphology, yielding a *PCC* of 0.835 and a *CS* of 0.828. The saturation issue with the original sensor, G_defIndBit, may explain why this correlation is not higher. Nonetheless, all tested sites demonstrated robust results, with *PCC* and CS consistently above 0.8. The posterior pulse and arm, boasting *PCC*s of 0.928 and 0.916, respectively, exhibited the highest correlations.

Analyzing the nEDs, the highest values corresponded to the default finger sensor paired with the low-gain sensor. Wrist and arm measurements displayed similar values, while lower limb sites (ankle and toe) exhibited higher distances.

Turning attention to the movement section of the signals (see Table 8), a reduction in values is evident, likely attributed to movement artifacts, a well-known noise source to which PPG sensors are susceptible. In this specific movement scenario, computer work, a smaller correlation reduction can be observed in the G_highArmSci and G_highPosSci signals.

### 4.4. Peak Detection

After comparing the ground truths with various signals in both scenarios (refer to Table 9 and Table 10), several conclusions can be drawn. In the at-rest data, the values for G_lowMidBit, both at 0.99, directly compare the ground truth created from this signal with its automatic detection. The nearly unitary values highlight the high precision of the automatic algorithm employed. Conversely, G_defIndBit values are lower, as anticipated, due to the higher saturation of the signal.

The original sensor, G_defIndBit, exhibits saturation specifically within the systolic peak of the PPG waveform. This saturation results in a waveform segment characterized by a persistent ’plateau’ of constant signal, deviating from the typical peak morphology, a global maximum followed by a decreasing phase. The absence of the correct morphology poses challenges for algorithms in accurately identifying the systolic peak, leading to diminished sensitivity and precision in peak detection.

In general, sensors in the new measurement sites demonstrate high precision and sensitivity in systolic peak detection, with the posterior wrist performing the best, boasting a mean sensitivity of 0.95 and mean precision of 0.94. However, analyzing the movement data reveals that all signals were rendered unusable for peak detection, with sensitivities hovering near 0.

### 4.5. Heart Rate Extraction

After assessing peak detection precision, HR difference, and percentual error, tables were generated for both at-rest and movement segments (refer to Table 11 and Table 12). Notably, G_lowMidBit exhibited excellent results compared to G_defIndBit, due to both sensors being in a similar anatomical location, with G_lowMidBit displaying a higher saturation percentage. In the new measurement sites, G_highArmSci, G_highPosSci, and G_highAntSci also demonstrated very low HR differences, all below 2 BPM. However, G_highAnkSci, with a mean HR difference of 7 BPM, exhibited a considerably higher value.

During movement periods, overall HR differences tended to be higher than during at-rest periods. Examining the percent error of various sensors, G_lowMidBit displayed an error of only 0.361%, while all other sensors, except for G_highAnkSci, maintained a percent error below 5%. This threshold is commonly used to assess acceptable error in clinical devices [66].

Poincaré graphics were also generated for each signal (see Figure 10, Figure 11, Figure 12, Figure 13, Figure 14, Figure 15, Figure 16 and Figure 17). Healthy shapes and sizes, resembling a comet symmetric with the identity line [67], were observed in various graphics from each sensor. This indicates that heart rate variability could be accurately studied using these signals.

## 5. Discussion

The main objectives of this work were twofold: first, to address the saturation issue observed in the original PulseSensor PPG sensor, and second, to evaluate the feasibility of alternative measurement sites for obtaining clear PPG signals.

To address saturation, we conducted a comparative analysis between the original sensor (G_defIndBit) and a proposed lower gain version (G_lowMidBit). While the mean saturation percentage exhibited a significant decrease, it did not reach zero. Examination of *SE* values indicated noteworthy changes only in the original sensor (0.032 and 0.043 during at-rest and movement periods, respectively). This suggests that the newly fine-tuned sensors experience minimal overall saturation, which is insufficient to distort the main frequency peak of the signals.

Although there are discernible differences in the original signal, they are not pronounced because saturation primarily occurs around the peak of a clean PPG, representing a small portion of the overall signal. However, even this minor difference in saturation can prevent accurate feature extraction. Notably, wave morphology maintained a high correlation, and systolic peak detection showed significant improvement. Furthermore, heart rate extraction demonstrated much greater precision.

Regarding the new measurement sites, both anterior and posterior wrist locations exhibited no significant saturation, displayed good signal quality based on spectral entropy, and maintained favorable wave morphology when compared to the G_lowMidBit signal. The posterior sensor yielded slightly superior results compared to the anterior sensor across various measures, particularly in detecting systolic peaks and calculating HR differences. Consequently, the posterior wrist emerges as the preferred region for wrist PPG data collection. This preference aligns with prior research supporting the use of the posterior wrist as a measurement site [68,69,70,71], with consistent findings [66,72].

Similarly, the upper arm site demonstrated values comparable to the wrist locations, justifying its use in blood pressure estimation [73]. In contrast, the ankle sensor exhibited poorer results, characterized by higher spectral entropy, lower wave morphology similarity, and reduced sensitivity to systolic peaks. Nevertheless, signal results were still relatively high, suggesting potential resolution through adjustments to the sensor’s gain or signal processing. Past studies have also explored this area for heart rate and blood pressure extraction, highlighting its potential [34,74].

Results from movement segments of the signals underscore motion artifacts as the primary challenge for PPG sensors in daily life recordings. Spectral entropy increased during these segments, yet wave morphology comparisons remained consistently high. This anomaly may be attributed to comparisons made among signals affected by similar movement artifacts, maintaining similar wave morphologies but no longer representing a clean PPG waveform. This observation is reinforced by the notably low sensitivities and precisions in systolic peak detection. Despite prior proposals suggesting the upper arm as a more motion artifact-resistant measurement site [75], our analysis of collected motion data did not corroborate this proposition.

## 6. Conclusions

In summary, our research successfully demonstrated that an alternative PPG sensor built on open-source hardware, readily available in the commercial market, is capable of collecting signals of superior quality. This study contributes to advancing the current state of the art by conducting a comprehensive analysis of innovative PPG sensors designed for various body locations and novel anatomical sites. Specifically, we found that reducing the default sensor’s gain effectively mitigated saturation issues and improved its ability to capture high-quality signals. Also, when equipped with suitable sensors, the posterior wrist and arm can yield readings of comparable quality to the established gold standard for PPG data collection, i.e. fingertip measurements.

## 7. Future Work

Limitations of this study are the restricted age range and the number of volunteers from whom data were collected. It is well-established that peripheral arterial circulation undergoes changes with age, primarily attributed to increased arterial stiffness [76]. This physiological shift may impact or distort PPG signals when derived from different anatomical locations [77]. Age has also been identified as an independent predictor of the second derivative plethysmogram wave [78].

Given these considerations, it is crucial to recognize the potential influence of age on signal quality and waveform characteristics. Future investigations should encompass a broader range of age groups to ensure that the newly proposed measurement sites and sensors maintain consistent signal quality across diverse age scenarios. This approach would contribute to a more comprehensive understanding of the generalizability and robustness of the findings. Nevertheless, the study herein described constitutes a fundamental prerequisite and an important stepping stone for these subsequent studies to be developed.

The saturation of the original sensor was not entirely eliminated, reason for which further small adjustments can be implemented to completely rectify this issue. Additionally, exploring the possibility of increasing the gain for the sensor adapted to ankle measurements may yield improved results.

In future work, it would be advantageous to incorporate a quality metric that evaluates not only the similarity of wave morphology but also its accuracy in relation to an optimal waveform.

## Figures and Tables

**Figure 1 sensors-24-00214-f001:**
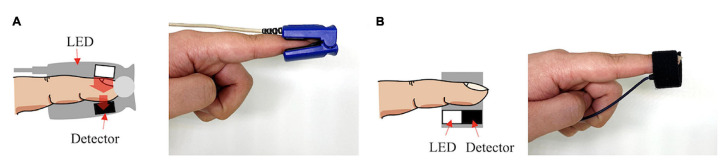
Transmissive PPG on the left (**A**), reflective PPG on the right (**B**). Extracted from [10].

**Figure 2 sensors-24-00214-f002:**
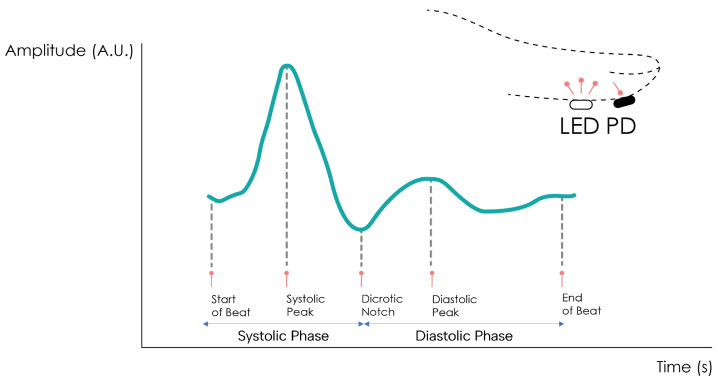
Normal PPG waveform of the CCC component. Adapted from [22].

**Figure 3 sensors-24-00214-f003:**
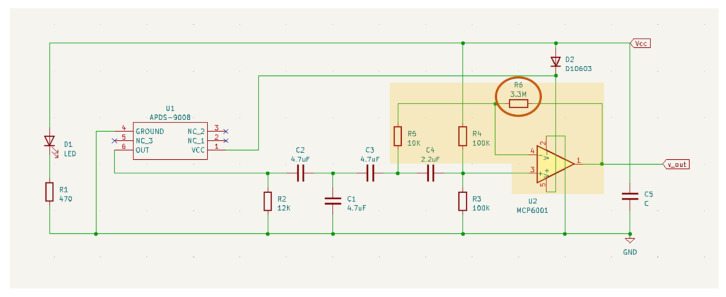
PulseSensor circuit schematics. Extracted from [56].

**Figure 4 sensors-24-00214-f004:**
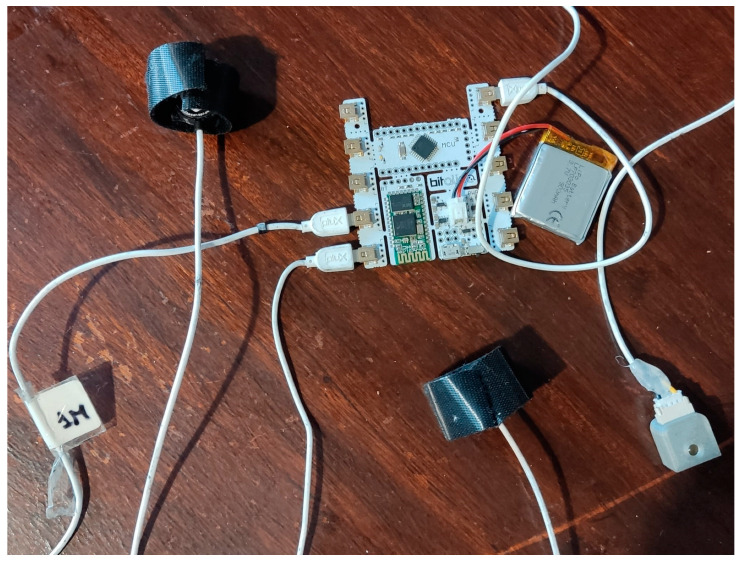
BITalino microprocessor and attached sensors. From left to right: G_low PPG sensor, G_def PPG sensor, LED.

**Figure 5 sensors-24-00214-f005:**
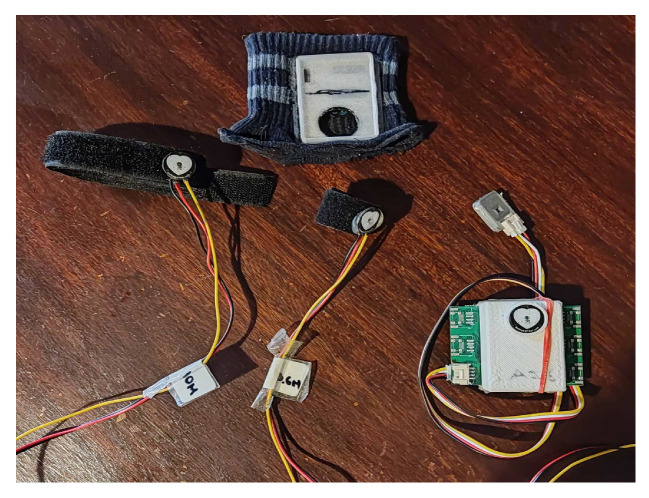
From left to right: G_high PPG sensor, G_def PPG sensor, ScientISST microprocessor inside 3D printed box with G_high PPG sensor inserted and Lux connected. Above the sensors is the elastic band with the 3D-printed socket attached.

**Figure 6 sensors-24-00214-f006:**
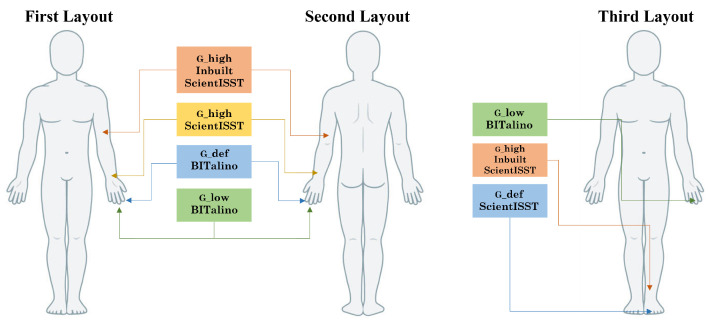
Diagram of the sensor placement in each of the layouts. The first and second layouts correspond to the anterior and posterior wrist positions, respectively.

**Figure 7 sensors-24-00214-f007:**
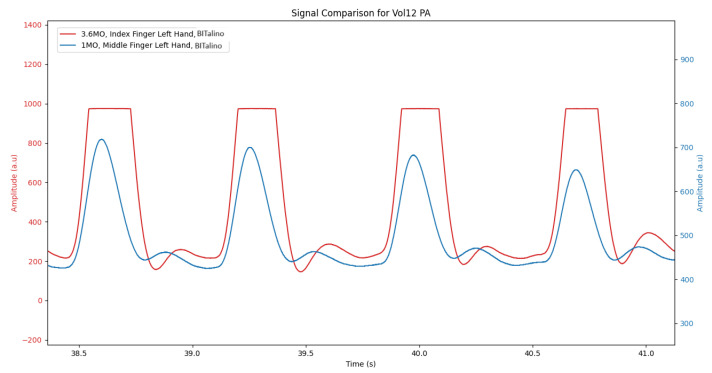
Raw signal comparison between sensors G_defIndBit (in Red) and G_lowMidBit (in Blue).

**Figure 8 sensors-24-00214-f008:**
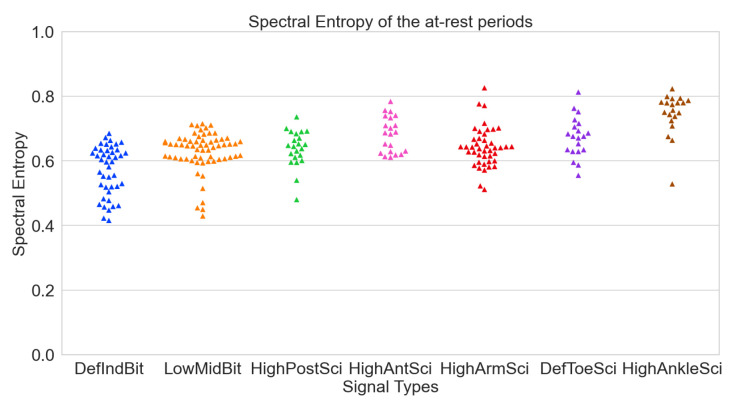
Spectral entropies of the various signals per sensor in at-rest periods.

**Figure 9 sensors-24-00214-f009:**
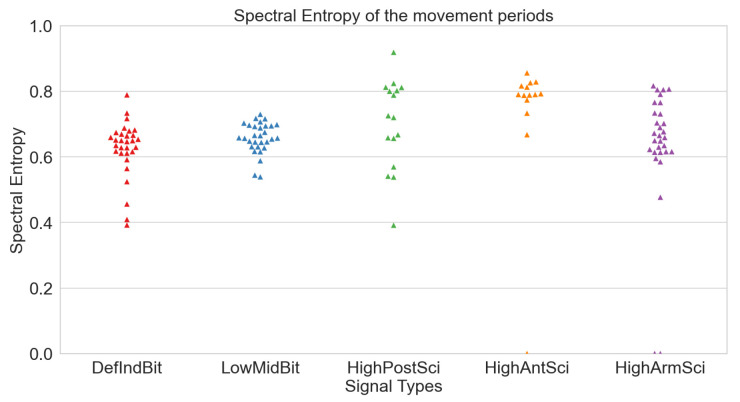
Spectral entropies of the various signals per sensor in movement periods.

**Figure 10 sensors-24-00214-f010:**
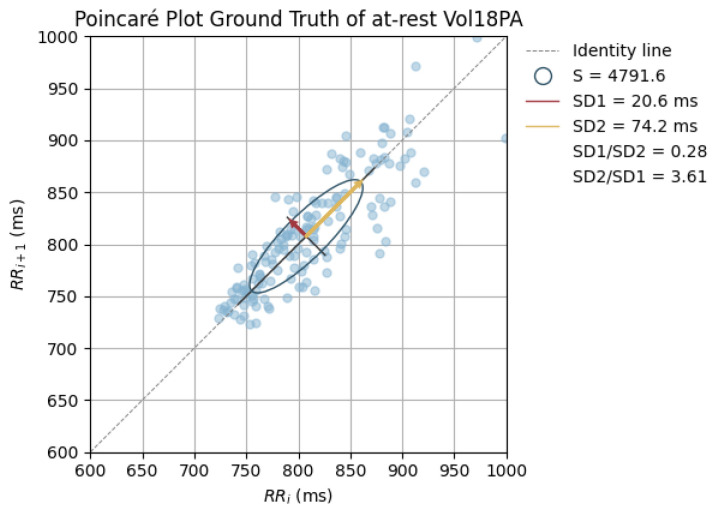
Poincaré graph of the ground truth at rest.

**Figure 11 sensors-24-00214-f011:**
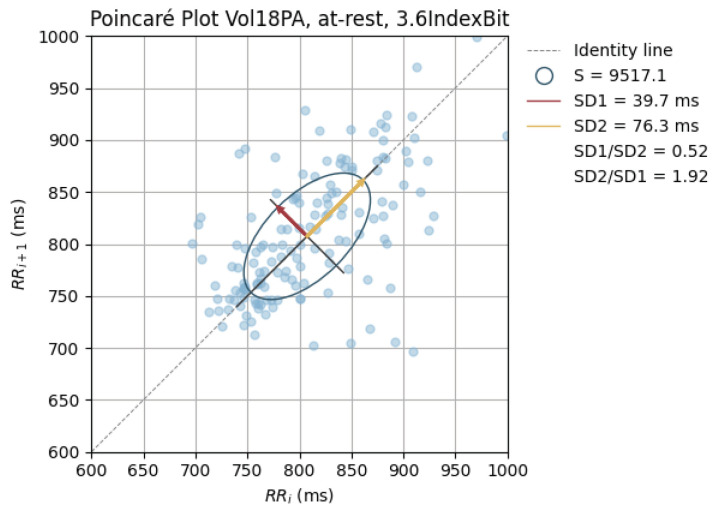
Poincaré Graph of the G_defIndBit at rest.

**Figure 12 sensors-24-00214-f012:**
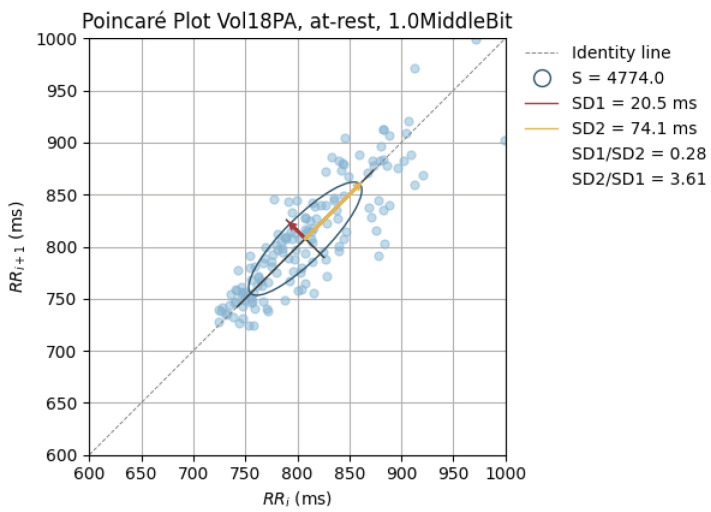
Poincaré graph of the G_lowMidBit at rest.

**Figure 13 sensors-24-00214-f013:**
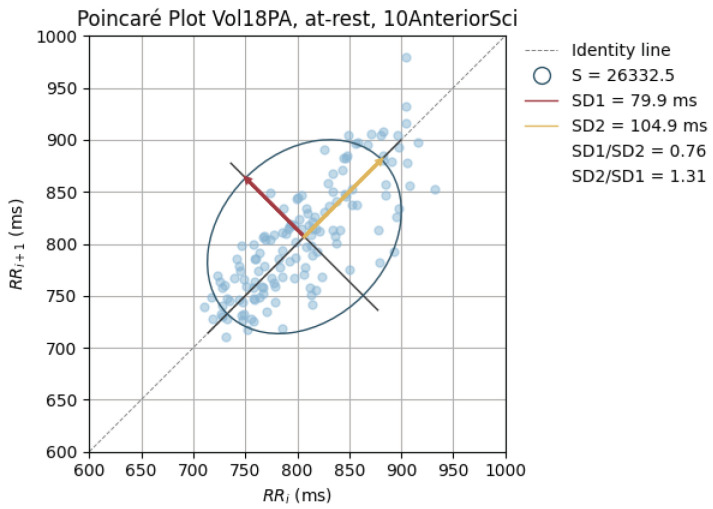
Poincaré graph of the G_highAntSci at rest.

**Figure 14 sensors-24-00214-f014:**
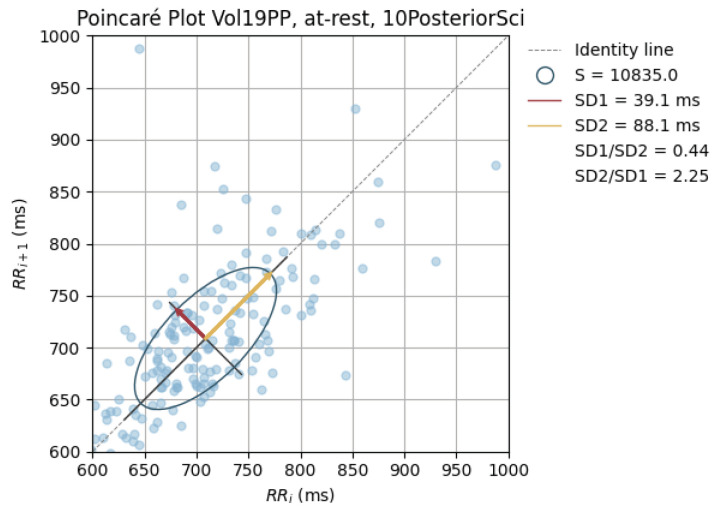
Poincaré graph of the G_highPosSci at rest.

**Figure 15 sensors-24-00214-f015:**
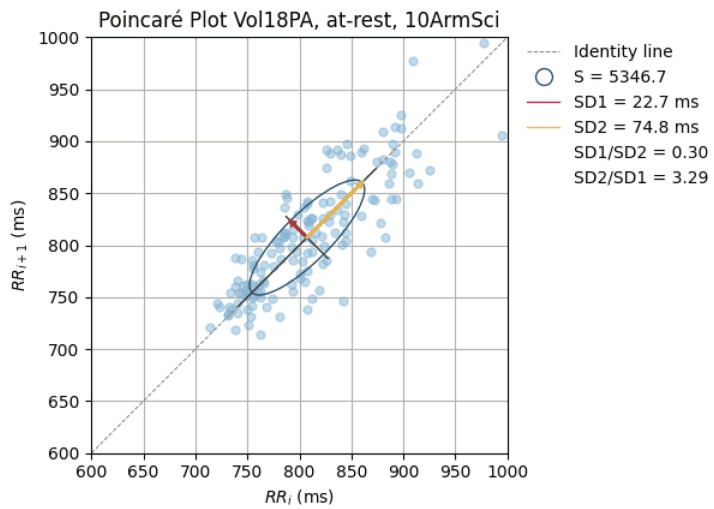
Poincaré graph of the G_highArmSci at rest.

**Figure 16 sensors-24-00214-f016:**
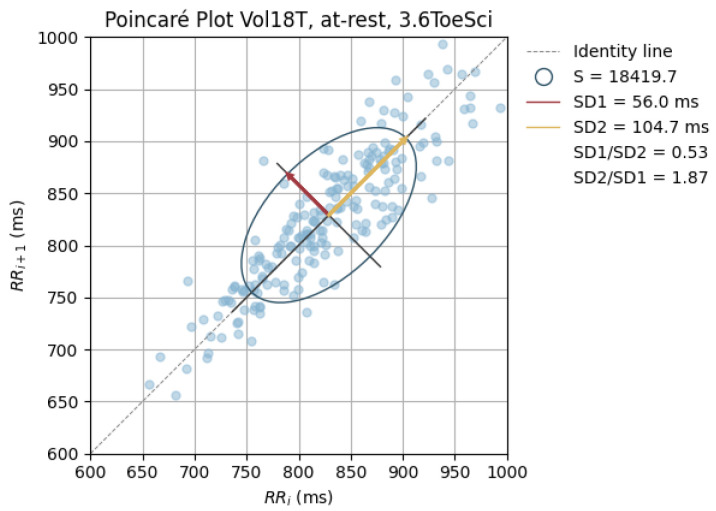
Poincaré graph of the G_defToeSci at rest.

**Figure 17 sensors-24-00214-f017:**
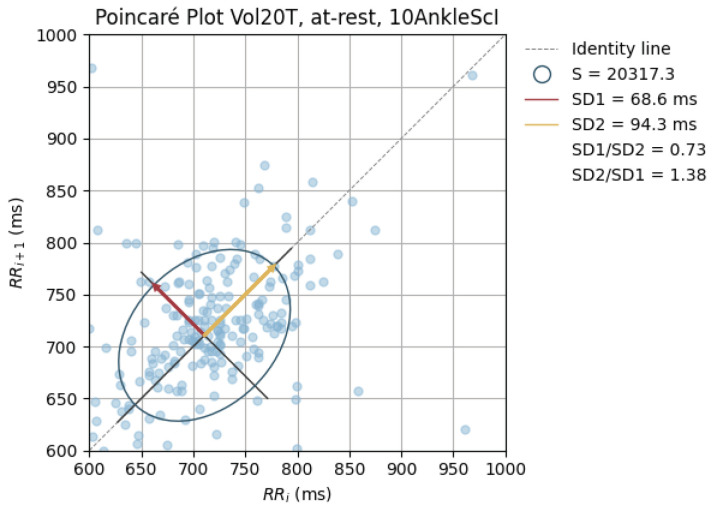
Poincaré graph of the G_highAnkSci at rest.

**Table 1 sensors-24-00214-t001:** Summary of sensor abbreviations, their *R6* resistance value, sensor type, the anatomical site, the device they are connected to, and the sample range of the signal.

Abreviation	*R6* (MΩ)	Sensor Type	Anatomical Site	Device	Sample Range
G_lowMidBit	1	G_low	Left Hand’s Middle Finger	BITalino	[0–1023]
G_defIndBit	3.3	G_def	Left Hand’s Index Finger	BITalino	[0–1023]
G_highArmSci	10	G_high	Left Anterior Arm	ScientISST	[0–8,388,607]
G_highPosSci	10	G_high	Left Posterior Wrist	ScientISST	[0–8,388,607]
G_highAntSci	10	G_high	Left Anterior Wrist	ScientISST	[0–8,388,607]
G_defToeSci	3.3	G_def	Left Foot’s Second Toe	ScientISST	[0–8,388,607]
G_highAnkSci	10	G_high	Left Medial Ankle	ScientISST	[0–8,388,607]

**Table 2 sensors-24-00214-t002:** Volunteers’ anthropometric data. The values are shown as median (min, max). For age, age ranges are used.

Parameter	All Subjects, *n* = 28
Age	[18–29] (18 to 59)
Height, m	1.69 (1.6 to 1.86)
Weight, kg	64.5 (45 to 100)
Body Mass Index (BMI), kg/m${}^{2}$	22.19 (15.94 to 30.86)

**Table 3 sensors-24-00214-t003:** Statistical sample values for saturation in different gain sensors.

Sensor Type	Min	Max	Median	Mean	N° of Samples
G_defBit	901	977	975	974 ± 3	14,396
G_lowBit	901	978	976	975 ± 8	13,939
G_defSci	8,001,720	8,298,304	8,061,480	8,061,040 ± 7681	13,624
G_highSci	8,001,900	8,272,080	8,208,360	8,205,810 ± 11,772	11,556

**Table 4 sensors-24-00214-t004:** Percentage of saturated signal in each sensor type and measurement site.

Sensor	Mean Saturation (%)	N° of Signals
G_lowMidBit	0.479 ± 1.294	68
G_defIndBit	18.554 ± 11.871	43
G_highArmSci	0.474 ± 1.210	43
G_highPosSci	<0.001 ± 0	22
G_highAntSci	<0.001 ± 0	21
G_defToeSci	<0.001 ± 0	20
G_highAnkSci	0.100 ± 0.331	20

**Table 5 sensors-24-00214-t005:** Mean spectral entropy for each signal type in at-rest periods.

Sensor	Mean *SE*	Mean *SE* without Saturation	N° of Signals
G_lowMidBit	0.629 ± 0.061	0.629 ± 0.061	43
G_defIndBit	0.572 ± 0.077	0.605 ± 0.056	43
G_highArmSci	0.643 ± 0.061	0.644 ± 0.061	43
G_highPosSci	0.639 ± 0.057	0.639 ± 0.057	22
G_highAntSci	0.683 ± 0.055	0.683 ± 0.055	21
G_defToeSci	0.674 ± 0.628	0.674 ± 0.063	20
G_highAnkSci	0.747 ± 0.066	0.746 ± 0.065	20

**Table 6 sensors-24-00214-t006:** Mean spectral entropy for each signal type in movement periods.

Sensor	Mean *SE*	Mean *SE* without Saturation	N° of Signals
G_lowMidBit	0.658 ± 0.047	0.657 ± 0.046	43
G_defIndBit	0.626 ± 0.086	0.669 ± 0.100	43
G_highArmSci	0.637 ± 0.191	0.634 ± 0.190	43
G_highPosSci	0.702 ± 0.138	0.702 ± 0.138	22
G_highAntSci	0.733 ± 0.216	0.733 ± 0.216	21

**Table 7 sensors-24-00214-t007:** Signal correlation metrics between different sensors and different measurement sites for the at-rest periods. Three parameters were used: the Pearson Correlation Coefficient, the Cosine Similarity, and the Normalized Euclidean Distance.

Sensor Pair	*PCC*	CS	nED	N° of Signals
G_lowMidBit/ G_defIndBit	0.835 ± 0.094	0.828 ± 0.088	5.438 ± 2.182	43
G_lowMidBit/ G_highArmSci	0.916 ± 0.055	0.914 ± 0.056	3.892 ± 1.269	43
G_lowMidBit/ G_highAntSci	0.911 ± 0.071	0.909 ± 0.071	3.810 ± 1.670	21
G_lowMidBit/ G_highPosSci	0.928 ± 0.051	0.927 ± 0.052	3.614 ± 1.367	22
G_lowMidBit/ G_defToeSci	0.879 ± 0.079	0.875 ± 0.080	4.892 ± 1.823	20
G_defToeSci/ G_highAnkSci	0.808 ± 0.131	0.803 ± 0.130	5.578 ± 2.331	20

**Table 8 sensors-24-00214-t008:** Signal correlation metrics between different sensors and different measurement sites for the movement periods. Three parameters were used: the Pearson Correlation Coefficient, the Cosine Similarity, and the Normalized Euclidean Distance.

Sensor Pair	*PCC*	CS	nED	N° of Signals
G_lowMidBit/ G_defIndBit	0.840 ± 0.120	0.832 ± 0.116	5.185 ± 2.457	43
G_lowMidBit/ G_highArmSci	0.879 ± 0.095	0.875 ± 0.095	4.635 ± 2.113	43
G_lowMidBit/ G_highAntSci	0.807 ± 0.155	0.799 ± 0.157	5.983 ± 3.039	21
G_lowMidBit/ G_highPosSci	0.860 ± 0.115	0.855 ± 0.116	5.034 ± 2.274	22

**Table 9 sensors-24-00214-t009:** Mean sensitivity and mean precision of peak detection using Elgendi’s algorithm for at-rest periods.

Sensor	Sensitivity	Precision	N° of Signals
G_lowMidBit	0.99 ± 0.01	0.99 ± 0.01	43
G_defIndBit	0.72 ± 0.25	0.71 ± 0.26	43
G_highArmSci	0.89 ± 0.22	0.89 ± 0.22	43
G_highPosSci	0.95 ± 0.11	0.94 ± 0.12	22
G_highAntSci	0.84 ± 0.26	0.84 ± 0.26	21
G_defToeSci	0.81 ± 0.30	0.78 ± 0.30	20
G_highAnkSci	0.74 ± 0.19	0.72 ± 0.20	20

**Table 10 sensors-24-00214-t010:** Mean sensitivity and mean precision of peak detection using Elgendi’s algorithm for movement periods.

Sensor	Sensitivity	Precision	N° of Signals
G_lowMidBit	0.019 ± 0.019	0.020 ± 0.020	43
G_defIndBit	0.017 ± 0.017	0.017 ± 0.017	43
G_highArmSci	0.018 ± 0.018	0.018 ± 0.018	43
G_highPosSci	0.020 ± 0.020	0.020 ± 0.021	22
G_highAntSci	0.018 ± 0.016	0.018 ± 0.016	21

**Table 11 sensors-24-00214-t011:** Mean heart rate difference in beats per minute and percent error between each sensor and the respective ground truth in the at-rest periods.

Sensor	Mean HR Difference (BPM)	Percent Error (%)	N° of Signals
G_lowMidBit	0.241 ± 0.557	0.361 ± 0.848	43
G_defIndBit	3.005 ± 6.771	4.620 ± 11.180	43
G_highArmSci	1.268 ± 3.978	1.848 ± 6.017	43
G_highPosSci	1.183 ± 2.993	1.832 ± 5.039	22
G_highAntSci	1.892 ± 1.968	2.669 ± 3.007	21
G_defToeSci	1.491 ± 3.019	2.138 ± 4.441	20
G_highAnkSci	7.076 ± 9.258	10.820 ± 15.894	20

**Table 12 sensors-24-00214-t012:** Mean heart rate difference in beats per minute and percentual error between each sensor and the respective ground truth in the movement periods.

Signal	Mean HR Difference (BPM)	Percent Error (%)	N° of Signals
G_lowMidBit	0.811 ± 1.325	1.067 ± 1.692	43
G_defIndBit	1.938 ± 2.793	2.640 ± 4.134	43
G_highArmSci	2.426 ± 4.293	3.418 ± 6.631	43
G_highPosSci	4.575 ± 5.017	6.279 ± 7.212	22
G_highAntSci	7.200 ± 6.193	9.421 ± 8.396	21

## Data Availability

Data are contained within the article.

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
