# Peer review of "Benchmarking of Sensor Configurations and Measurement Sites for Out-of-the-Lab Photoplethysmography"

_sensors, 2023, doi:10.3390/s24010214_

Round 1
Reviewer 1 Report
Comments and Suggestions for Authors
This study modified an off-the-shelf PPG sensor to improve signal saturation. The feasibility of using an optimized sensor in the lower limb as an alternative measurement site was explored. Data from 28 participants aged 18 to 59 years was collected. The adjusted PPG sensors provide high-quality signals in the fingertips, and the posterior pulse and upper arm can extract high-quality signals as well. I have some suggestions that the paper may be improved.
Including information about the IAC and IDC parts of the saturations may improve the understanding of PPG signals in section 2.1, Photoplethysmography. This will provide a better understanding of why the intensity of the captured light depends on these factors.
In section 3.3, Quality Evaluation, there are extra blank paragraphs. You may rearrange the entire area.
Please note that in Table 2, the median value for the Age parameter is provided as a range. Kindly review it and verify the values.
It is unclear where the equation on line 178 is applied.
In Figure 8, yellow dots are not clearly seen.
Figure 10 is not clearly visible. Please increase the figure size or use AI image processing tools such as stable diffusion for image quality improvement.
In Figure 6, you may indicate which representation is anterior and posterior.
Reviewer 2 Report
Comments and Suggestions for Authors
Proposed study aimed to address saturation issues in the original PulseSensor PPG sensor and assess the viability of new measurement sites for obtaining clean PPG signals. The sensor is modified to reduce saturation and is evaluated for performance in both upper (index finger) and lower limb (first toe) locations. Overall, the modified PPG sensors offer a promising alternative for obtaining high-quality signals in both traditional and novel measurement sites.
It is recommended that the following issues be detailed in the study carried out:
1. In the study, BITalino and ScientISST CORE1 devices were used, but a more comprehensive comparison between these devices was not conducted. The advantages, disadvantages, and performance characteristics of each device should be detailed. Additionally, the reasons for choosing these specific devices should be explained.
2. The article discusses efforts to reduce signal saturation by changing the resistance values in the sensor amplification stage. However, it should be discussed whether there are other optimization strategies related to sensor design and structure. If such strategies exist, what they are and how they can be implemented should be explored.
3. The article uses a specific PPG sensor, but comparing the performance of different PPG sensors could add value to the study. The advantages and disadvantages of sensors from different brands and models should be examined, and the rationale behind choosing the particular sensor used in the study should be explained.
4. The article evaluates new measurement points, but a more detailed explanation of the advantages and limitations of PPG signals obtained from these points, supported by relevant literature, should be provided.
5. The study has a limited number of participants and includes individuals within a specific age range. A discussion about the implications of this limitation and its potential impact on the generalizability of the results would be appropriate.
6. The writing and presentation of the study could be made more understandable. Specifically, the methodology, results, and discussion sections could be articulated in a clearer manner.
Reviewer 3 Report
Comments and Suggestions for Authors
– small errors in the text, missing spaces before the beginning of the sentence, etc.
l.26 - The word "biometrics" is not suitable in this context. Biometrical data of the patient may be transfered too, but biometrics is not the state that needs monitoring.
– the abbreviation PPG is not defined in the Abstract, on the other hand, it is redefined in the 1st sentence of 2.1 (l.51) vs Introduction (l.31).
– References to figures 4 and 5 (l.104) are before the reference to Fig.3.
– The item [35] in line 108 has no internal link to references and the reference [35] in the list of references has no external link to https://www.afonso-ferreira.net/projects/tuning-pulse-sensor
– Tab2 “Body Mass Index (BMI), kg/m” – must be calculated as kg/m^2
– l.178 - “the mean value subtracted by one standard deviation (mean - 1xStD)” – unusual acronym, msut be explained why 1x and no 2x or 3x.
– l.248 - here it is denoted OK
– l.187 – missing relation for entropy calculation and specification of the entropy, whether it is Shannon entropy, or Renyi entropy or Tsallis entropy
– l.229 – redefinition of the abbreviation HR
– In Figs. 8,9 all the text and numbering of axes have very small size.
– Table 6. – only little differences in SE for saturated a clean PPG signals
– Tables 7,8 – Why are the values so different for the normalized Euclidean norm?
– Tables 9,10 – Why are the results the worst for the sensors G_defIndBit?
– Fig. 10 – totally unnreadable text and numbering of axes as well as legens => rearrange better in two pairs in two rows and greater !
– l.312-360 – “Discussion & Conclusion” – At first glance it seems that it contains only discussion – The text should be better structured and divided, for example plans for future or what to do with it further (usage for others).
– In the reference list, there are some lex. mistakes – e.g. [1] missing dot “.” before “Available online:” dtto [34], [35]
– Ref. [31] has 2 times the year 2023, dtto [47]
– Ref. [39] “2015-.”
